# Magnetic skyrmion braids

Fengshan Zheng [1✉], Filipp N. Rybakov[2✉], Nikolai S. Kiselev [3✉], Dongsheng Song[1,4], András Kovács[1], Haifeng Du [5], Stefan Blügel [3] & Rafal E. Dunin-Borkowski[1]

Skyrmions are vortex-like spin textures that form strings in magnetic crystals. Due to the analogy to elastic strings, skyrmion strings are naturally expected to braid and form complex three-dimensional patterns, but this phenomenon has not been explored yet. We found that skyrmion strings can form braids in cubic crystals of chiral magnets. This finding is confirmed by direct observations of skyrmion braids in B20-type FeGe using transmission electron microscopy. The theoretical analysis predicts that the discovered phenomenon is general for a wide family of chiral magnets. These findings have important implications for skyrmionics and propose a solid-state framework for applications of the mathematical theory of braids.

[1] Ernst Ruska-Centre for Microscopy and Spectroscopy with Electrons and Peter Grünberg Institute, Forschungszentrum Jülich, Jülich, Germany. [2] Department of Physics, KTH-Royal Institute of Technology, Stockholm, Sweden. [3] Peter Grünberg Institute and Institute for Advanced Simulation, Forschungszentrum Jülich and JARA, Jülich, Germany. [4] Information Materials and Intelligent Sensing Laboratory of Anhui Province, Key Laboratory of Structure and Functional Regulation of Hybrid Materials of Ministry of Education, Institutes of Physical Science and Information Technology, Anhui University, Hefei, China. [5] The Anhui Key Laboratory of Condensed Matter Physics at Extreme Conditions, High Magnetic Field Laboratory, Chinese Academy of Science (CAS), Hefei, China. ✉email: f.zheng@fz-juelich.de; prybakov@kth.se; n.kiselev@fz-juelich.de

Filamentary textures can take the form of braided, rope-like superstructures in nonlinear media such as plasmas and superfluids[1–3]. The formation of similar superstructures in solids has been predicted, for example, from flux lines in superconductors[4]. However, their observation has proved challenging[5,6].

Here, we provide unambiguous experimental evidence for braids of skyrmion strings in cubic crystals of chiral magnets.

Magnetic skyrmions[7–10] have been observed over a wide range of temperatures and applied magnetic fields in Ge-based and Si-based alloys with B20-type crystal structures, such as FeGe[11–14], MnSi[15], Fe$_{1-x}$Co$_x$Si[16,17] and others[18]. These skyrmions are topologically nontrivial textures of the magnetization unit vector field $\mathbf{m}(\mathbf{r})$, whose filamentary structures resemble vortex-like strings or tubes, with typical diameters of tens of nanometers. The length of such a string is assumed to be limited only by the shape and size of the sample[19], with direct observations confirming the formation of micrometer-long skyrmion strings[20,21]. Previous studies have suggested that magnetic skyrmion strings are (nearly) straight[19–25]. In contrast, here we show that skyrmion strings are able to twist around one another to form superstructures that we refer to as *skyrmion braids* (Fig. 1).

We begin with a theoretical description, which is based on a micromagnetic approach and a standard model of cubic helimagnets (chiral magnets), such as FeGe (see Methods). Static equilibrium in such a system results from a delicate balance between Heisenberg exchange, Zeeman interaction, chiral Dzyaloshinskii–Moriya interaction[26,27] (DMI), and dipole–dipole interaction. The equilibrium state is typically a conical spin spiral, whose modulation axis is parallel to the applied field $\mathbf{B}_{ext}$ (inset to Fig. 1a). The period of such a spiral $L_D$ is nearly constant for each material, taking a value of ~70 nm for FeGe[28]. The conical phase can be regarded as a natural "vacuum" (background), in which clusters and isolated strings of skyrmions are embedded[23,24].

## Results

**Micromagnetic simulations.** We use energy minimization (see Methods) to show that interacting skyrmion strings can lower their total energy by twisting into braids. A representative example of such a skyrmion braid is shown in Fig. 1 for an energetically favorable configuration of five skyrmion strings, which wind around a sixth straight string at their center. Such a winding braid forms spontaneously from an initial configuration

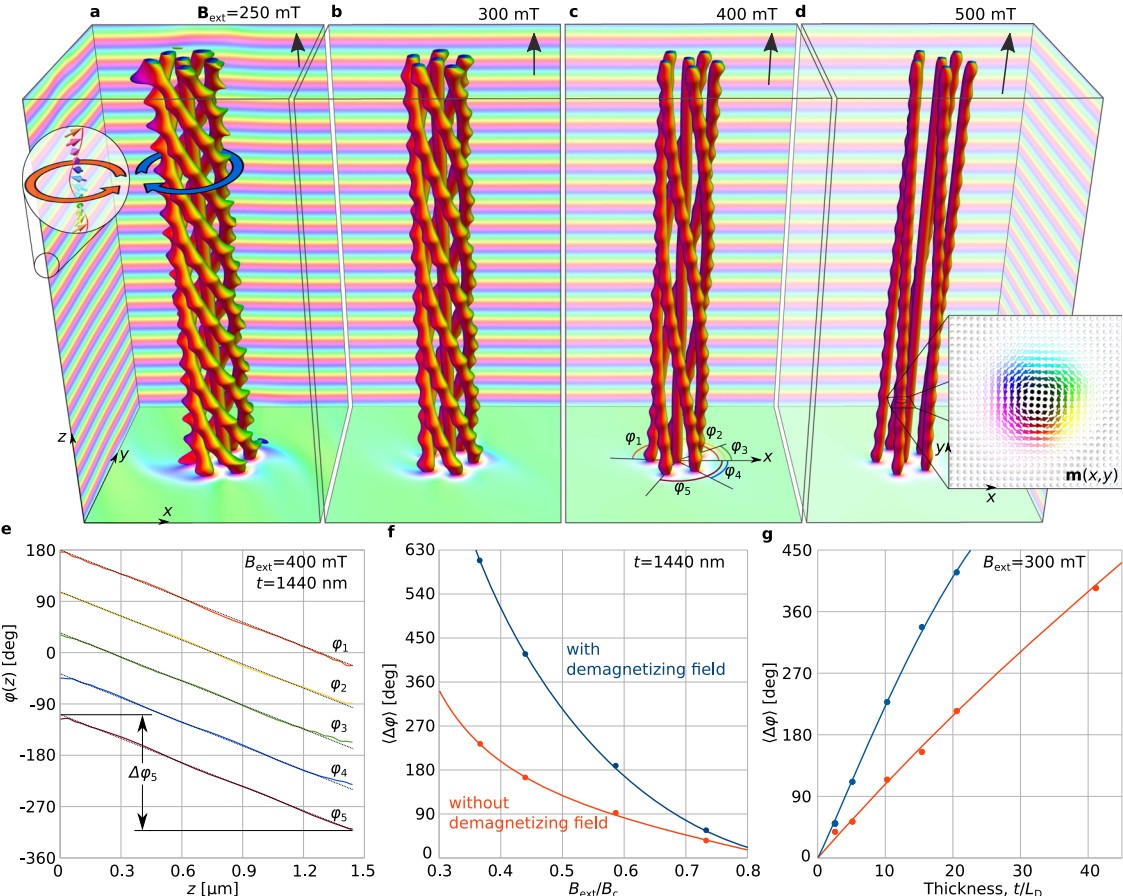

**Fig. 1 Skyrmion braid in a chiral magnet. a–d** Skyrmion braid comprising six skyrmion strings represented by isosurfaces of $m_z = 0$. The color modulation at the edges of the box indicates the presence of the conical phase. (see inset in **a**). The equilibrium state for each value of applied magnetic field ($\mathbf{B}_{ext} \| \hat{\mathbf{e}}_z$) is found by using energy minimization (see "Methods"), on the assumption of periodic boundary conditions in the $xy$ plane, free surfaces in the third dimension, and a sample thickness of $t = 1440$ nm. The presence of bumps on the isosurfaces results from the magnetization modulations in the surrounding conical phase. These bumps become less pronounced with increasing $B_{ext}$, as the cone phase approaches a field-polarized state. **e** Dependence of twist angle on distance to the lower surface for the five skyrmions that wind around a central sixth skyrmion. The angles $\varphi_i(z)$ are measured from the $x$ axis, as indicated in **c**. The dotted lines are linear fits for each $\varphi_i(z)$ dependence. **f** Dependence of the average twist angle $\langle \Delta\varphi \rangle = \frac{1}{5}\sum_i [\varphi_i(0) - \varphi_i(t)]$ on the applied magnetic field. **g** Dependence of the average twist angle $\langle \Delta\varphi \rangle$ on sample thickness $t$. The solid circles in **f**, **g** are obtained from numerical calculations, while the lines are used to guide the eye.

of six straight skyrmion strings (see Methods and Supplementary Movie 1). A skyrmion braid has one energetically preferable chirality, which depends on the sign of the DMI constant $D$. Without loss of generality, we assume below that $D > 0$. The chirality of the twist is opposite to that of the magnetization of the surrounding cone phase, as indicated using blue and red arrows, respectively, in Fig. 1a. Although this behavior may appear to be counterintuitive, along any line parallel to the $z$ axis, the spins have the same sense of rotation about $\hat{\mathbf{e}}_z$ as for the conical spiral. The twisting or untwisting of a skyrmion braid by an external magnetic field, as illustrated in Fig. 1a–d, is fully reversible (see Supplementary Movie 1).

In order to quantify the degree of twist, we trace the position of each skyrmion by its center, where the magnetization is antiparallel to $\mathbf{B}_{ext}$. We define the twist angle $\varphi_i(z)$ for the $i^{th}$ skyrmion in each $z$ plane as the angle between the $x$ axis and the line connecting the central and $i^{th}$ skyrmion (Fig. 1c). The $\varphi_i(z)$ curves shown in Fig. 1e are almost linear and have identical slopes, with small deviations close to the free surfaces due to the chiral surface twist[22]. A linear fit to each plot of $\varphi_i(z)$ can be used to estimate the total twist angle $\Delta\varphi_i$, by which the strings rotate from the lower to the upper surface.

Figure 1f shows the dependence of the average twist angle $\langle\Delta\varphi\rangle = \frac{1}{5}\sum_i \Delta\varphi_i$ on the applied magnetic field, both with and without consideration of demagnetizing fields, to illustrate the role of dipole–dipole interactions. The twist angle is reduced by a factor of ~2 when demagnetizing fields are neglected. This difference decreases with increasing $B_{ext}$. The applied magnetic field is shown in reduced units relative to the critical field $B_c$ in Fig. 1f, in order to facilitate a comparison between the results obtained with and without demagnetizing fields. In both cases, $B_c$ is the phase-transition field between conical and saturated ferromagnetic states (see Methods). Figure 1g shows the dependence of the twist angle $\langle\Delta\varphi\rangle$ on sample thickness $t$ at fixed $B_{ext}$. Although dipole–dipole interactions increase the degree of twist significantly, they are not necessary for the stability of skyrmion braids.

The quasilinear dependence of $\varphi_i(z)$ on $z$ in Fig. 1e and the monotonic dependence of $\langle\Delta\varphi\rangle(t)$ on $t$ in Fig. 1g suggest that the volume and the free surfaces both play important roles in the formation of skyrmion braids.

Calculations performed for a bulk sample with periodic boundary conditions show that, for example, in Fig. 2a, a six-string skyrmion braid is energetically more favorable than a cluster of straight strings in fields below $B^* \approx 0.61 B_c$ (see the vertical dashed line in Fig. 2b). Figure 2c illustrates the details of the energy redistribution for an arbitrarily chosen value of $B_{ext}$ below $B^*$. In a thin plate, however, the same braid is stable above $B^*$ (Fig. 1f), suggesting that surface effects may additionally enhance braid stability. The mechanism of skyrmion braid stability therefore results from a sophisticated interplay of multiple energy terms and depends on the geometry of the system. The dipole–dipole interaction does not play a fundamental role in the energy stability of a braid, but contributes to it. Instead, the stability of a braid results primarily from the energies of exchange-type interactions associated with texture distortions.

**Experimental observations.** The dependence of the average twist angle $\langle\Delta\varphi\rangle$ on plate thickness shown in Fig. 1g suggests that the helical twist of skyrmion strings should be observable in a thin sample, for which $t \sim L_D$. This criterion is important for observations using transmission electron microscopy (TEM), for which an electron-transparent sample is required. For $Fe_{0.5}Co_{0.5}Si$, with $L_D \sim 90$ nm, the thickness below which a sample is electron-transparent at an accelerating voltage of 300 kV is ~3.3 $L_D$,

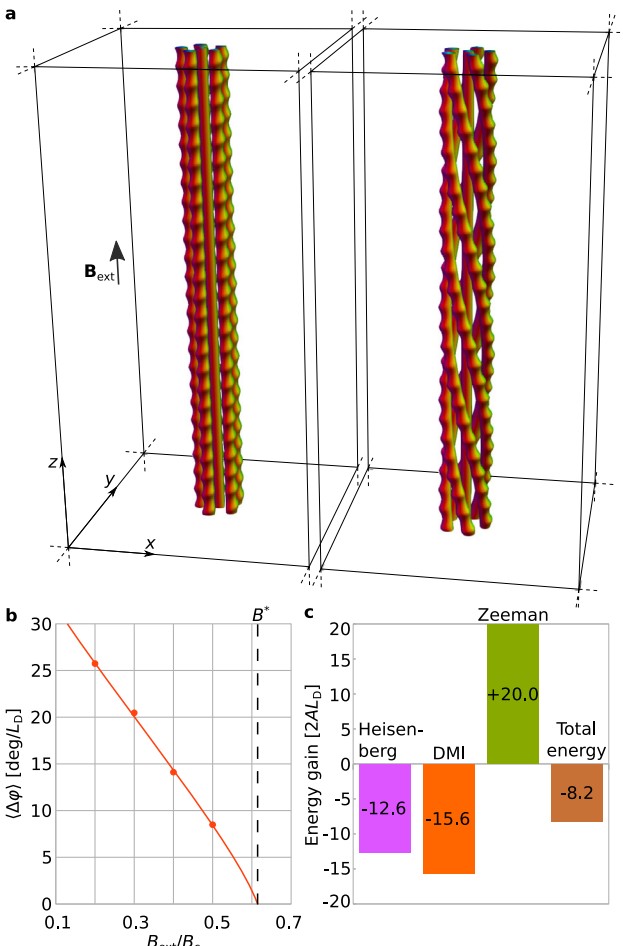

**Fig. 2 Energy gain of a skyrmion braid in a bulk sample. a** Cluster of six straight skyrmion strings (left) and a corresponding skyrmion braid (right) in a bulk sample with periodic boundary conditions in all three dimensions. **b** Dependence of energetically optimal twist per unit length on the applied magnetic field calculated without dipole–dipole interactions. **c** Component-wise energy difference between a cluster of straight skyrmion strings and a skyrmion braid at $B_{ext} = 0.392\ B_c$ for a period along $z$ equal to 25 $L_D$, with $\langle\Delta\varphi\rangle \approx 14\ \deg/L_D$.

corresponding to ~300 nm[17]. This value is assumed to be approximately the same for FeGe.

We studied three high-quality electron-transparent FeGe lamellae with thicknesses of ~180 nm (~2.6 $L_D$), which is large enough for the formation of skyrmion braids but small enough for magnetic imaging in the TEM. The lateral dimensions of the samples were $1 \times 1$ μm (S1), $800 \times 540$ nm (S2), $6 \times 7$ μm (S3). We observed skyrmion braids in all three samples. Below, we provide representative results from samples S1 and S2. Other results are included in Supplementary Figs. 2–9 and Movies 3–6.

For experimental observations of skyrmion braids, we designed the following protocol. First, we performed magnetization-reversal cycles until a desired number of skyrmion strings had nucleated in the sample. Multiple cycles were required due to the probabilistic character of skyrmion nucleation. In order to avoid the interaction of skyrmions with the edges of the sample, the strength of the external magnetic field was increased to move the skyrmions toward the center of the sample. We then gradually decreased the applied magnetic field until the contrast of the straight skyrmion strings changed to those expected for braids. (see Supplementary Movie 5). Although this is not a unique

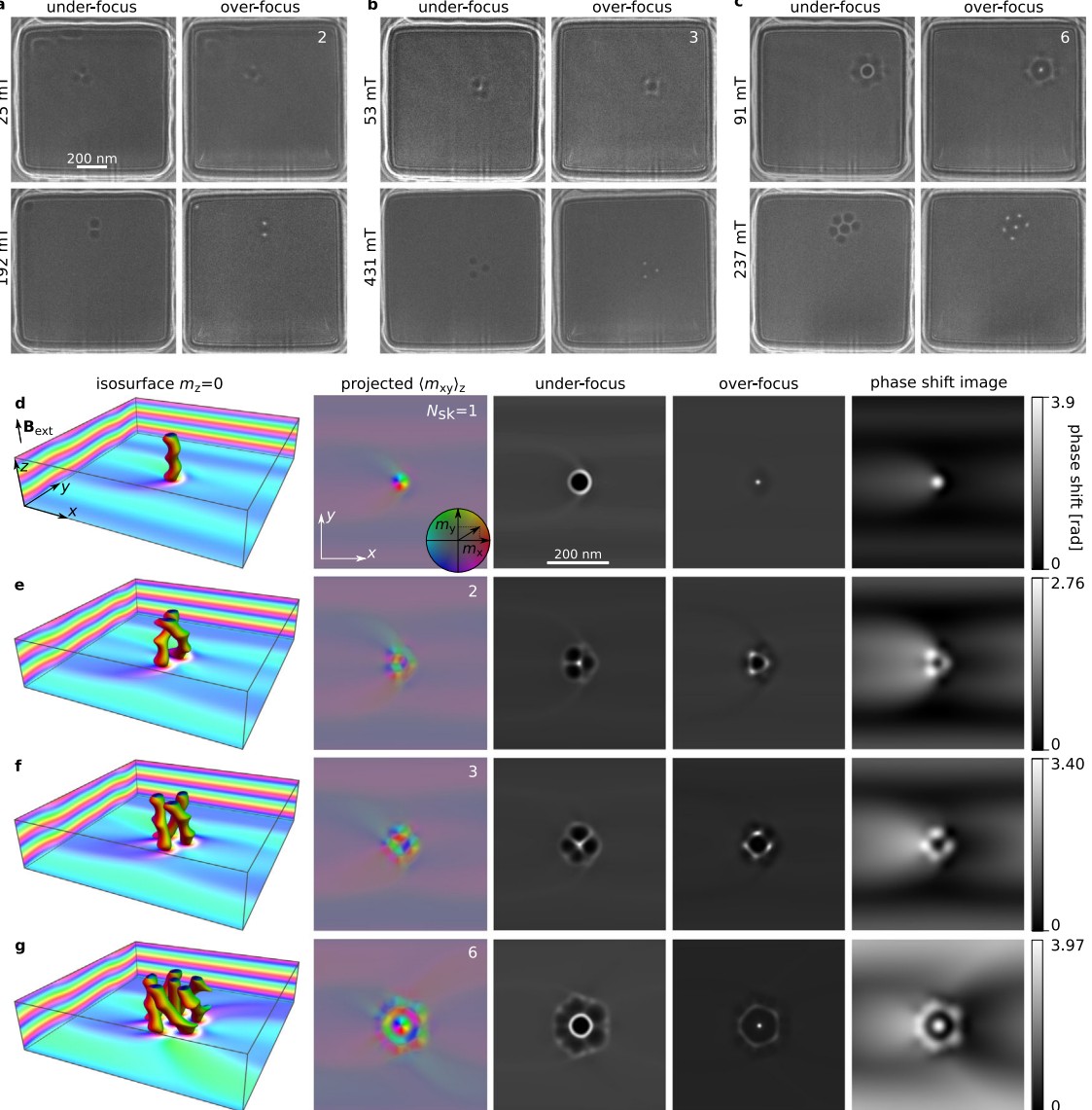

**Fig. 3 Skyrmion braids in an extended FeGe plate of thickness 180 nm. a–c** Experimental Lorentz TEM images of twisted skyrmions–skyrmion braids (upper row) and untwisted (lower row) skyrmions recorded at two different magnetic fields in sample S1 at 95 K. **d–g** Theoretical results for $B_{ext} = 275$ mT, assuming periodic boundary conditions in the $xy$ plane and free surfaces in the third dimension. From left to right, each row shows: an equilibrium state represented by isosurfaces of $m_z = 0$; in-plane magnetization $\langle \mathbf{m}_{xy} \rangle_z$ averaged over the thickness of the plate; underfocus and overfocus Lorentz TEM images; electron optical-phase image. The index at the top-right corner in **a–c** and in the simulated images of $\langle \mathbf{m}_{xy} \rangle_z$ indicates the number of skyrmion strings $N_{sk}$ in the braid. An isolated skyrmion string is shown in **d** for comparison with skyrmion braids in **e–g**.

protocol for nucleating skyrmion braids (see, e.g., Supplementary Fig. 3), it results in high reproducibility of the results. (See Supplementary Movies 4 and 6). We did not observe skyrmion braids during in-field cooling of the samples.

Figure 3 shows experimental Lorentz TEM images (**a–c**) and theoretical results (**d–g**) for short skyrmion braids. Here, we compare only the Lorentz TEM images. See Supplementary Fig. 10 for different defocus values and Supplementary Fig. 11 for the case of opposite chirality of the material. Representative experimental phase-shift images for sample S1 recorded using off-axis electron holography are provided in Supplementary Fig. 2. Good agreement is obtained between the experimental and simulated images. The key feature of these magnetic images of skyrmion braids is the appearance of a ring-like pattern and a significant reduction in contrast in comparison with straight skyrmions, as shown in the lower row in Fig. 3a–c. The

correlation between the magnetic images and average in-plane magnetization components $\langle \mathbf{m}_{xy} \rangle_z = t^{-1} \int \mathbf{m}_{xy} dz$ in Fig. 3d–g arises because, to a first approximation, the magnetic field follows the $\mathbf{m}$ vector. It should be noted that the electron beam is perpendicular to the plate and only sensitive to $xy$ component of the magnetic field within and around the sample. In sample S1 (and similarly in sample S3—see Supplementary Fig. 9), the typical applied fields that were found to stabilize braids were different from theoretical values. This discrepancy can be attributed to minor damage from the fabrication process[29], including the formation of thin amorphous magnetic surface layers[30–32]. In contrast, excellent agreement with theory was achieved for sample S2, which was also used in Ref. [33].

Figure 4 shows experimental and theoretical images for sample S2 (see also Supplementary Fig. 4). Figure 4a shows a three-string skyrmion braid. Figure 4b shows a skyrmion braid comprising six

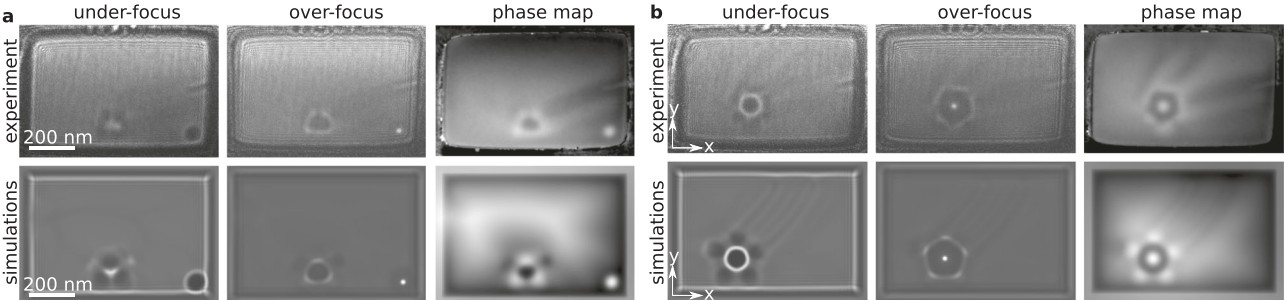

**Fig. 4 Comparison between experimental and simulated images of skyrmion braids. a, b** Magnetic images of skyrmion braids containing three and six skyrmion strings, respectively. The upper panel shows experimental underfocus and overfocus Lorentz TEM images and electron optical phase-shift images recorded using off-axis electron holography from sample S2 at 95 K. The lower panel shows the corresponding simulated Lorentz TEM images and phase-shift images. The perpendicular applied magnetic field in **a** and **b** is 218 and 187 mT, respectively. The high-contrast feature at the lower-right corner in **a** is an isolated skyrmion string attached to the edge. The sample size, external magnetic field, defocus and accelerating voltage are the same for the experimental and simulated images.

strings in a pentagonal pattern, similar to those shown in Fig. 3c, g. As a result of its smaller size, electron phase-shift images could be recorded from this sample using off-axis electron holography (see Methods). Excellent agreement is obtained between experiment and theory for both underfocus and overfocus Lorentz TEM images and electron phase-shift images, providing confidence in the interpretation of the experimental data. The skyrmion braids can be stabilized over a wide temperature range, as shown in the form of representative magnetic images in Supplementary Fig. 8 for sample S1 at 120 K and Supplementary Fig. 6 for sample S2 at 170 K.

## Discussion

Because of the magnetization inhomogeneities along the skyrmion braids, their electromagnetic properties are expected to differ from those of straight skyrmion strings. Electrons passing along the braids accumulate a Berry phase and experience emergent electric and magnetic fields that are modulated with the thickness of the sample and quantized by the number of skyrmion strings in the braid. This, in turn, gives rise to magnetoresistive and transport phenomena that would not be observed for straight skyrmion strings. A promising system for demonstrating these effects is a magnetic nanowire[34,35], in which a skyrmion braid can occupy the entire volume. In larger samples, skyrmion braids are expected to differ in structure and close-packing form[36,37]. See additional examples of superstructures of braided skyrmions in Supplementary Figs. 12 and 13.

In contrast to magnetic hopfions[38–40], skyrmion braids cannot be localized completely in three dimensions without singularities. In other words, skyrmion strings start on one surface of a sample and end on the other surface. A skyrmion braid represents a geometric braid[41] when the ends of the skyrmion strings are fixed on the sample surface due to pinning or surface engineering. It is then topologically protected and cannot be unwound. The latter behavior is guaranteed by the robustness of skyrmion strings, resembling strides[42] of a geometric braid. Such skyrmion braids can be described in terms of an Artin braid group[43] $\mathcal{B}_N$. In this sense, skyrmion braids are three-dimensional analogs of anyons[44] in $(2 + 1)$-dimensional space-time.

In summary, we have discovered novel magnetic superstructures, which we term skyrmion braids, using electron microscopy and micromagnetic calculations. The generality of our theoretical approach suggests that superstructures of skyrmion strings that wind around one another can be formed in all noncentrosymmetric cubic magnets, offering new perspectives for studies and applications of the anomalous (topological) Hall effect[34,45,46], magnetic resonance, spin waves on skyrmion strings[47,48], and magnetization dynamics driven by currents[34].

## Methods

**Micromagnetic calculations**. For a theoretical description of skyrmion braids, we follow the micromagnetic approach. In the most general case, when the demagnetizing field is taken into account, the state of a system is described in terms of two coupled vector fields: the magnetization field $\mathbf{M}(\mathbf{r})$, which is defined only inside the volume of the magnet $V_m$, and the magnetic field $\mathbf{B}(\mathbf{r})$, which is defined in all space. The field $\mathbf{B}(\mathbf{r})$ is typically the sum of a homogeneous applied magnetic field $\mathbf{B}_{ext}$ and the demagnetizing field produced by the magnetic sample itself. It can be expressed in the form

$$\mathbf{B} = \mathbf{B}_{ext} + \nabla \times \mathbf{A}_d, \qquad (1)$$

where $\mathbf{A}_d(\mathbf{r})$ is the component of magnetic vector potential due to the presence of the magnetization field $\mathbf{M}(\mathbf{r})$. The total energy of the system (up to an additive constant) is the sum of the exchange energy, the DMI energy, the Zeeman energy, and the self-energy of the demagnetizing field:

$$\mathcal{E} = \int_{V_m} d\mathbf{r}\, \mathcal{A}\, |\nabla \mathbf{m}|^2 + \mathcal{D}\, \mathbf{m} \cdot (\nabla \times \mathbf{m}) - M_s\, \mathbf{m} \cdot \mathbf{B}$$
$$+ \frac{1}{2\mu_0} \int_{\mathbb{R}^3} d\mathbf{r}\, |\nabla \times \mathbf{A}_d|^2, \qquad (2)$$

where $\mathbf{m}(\mathbf{r}) = \mathbf{M}(\mathbf{r})/M_s$ is a unit vector field that defines the direction of the magnetization, $M_s = |\mathbf{M}(\mathbf{r})|$ is the saturation magnetization, $\mathcal{A}$ is the exchange-stiffness constant, $\mathcal{D}$ is the constant of isotropic bulk DMI, and $\mu_0$ is the vacuum permeability ($\mu_0 \approx 1.256 \times 10^{-6}$ N A$^{-2}$). The notation $|\nabla \mathbf{m}|^2 \equiv \sum_{i=x,y,z} |\nabla m_i|^2$ denotes the Euclidean norm of gradients of $\mathbf{m}$. In our simulations, we used the following material parameters for FeGe[33]: $\mathcal{A} = 4.75$ pJm$^{-1}$, $\mathcal{D} = 0.853$ mJm$^{-2}$, and $M_s = 384$ kAm$^{-1}$.

Static equilibrium solutions discussed in the main text were obtained by numerical minimization of (2) for the pair of fields $\mathbf{m}$ and $\mathbf{A}_d$. Such an approach for the solution of magnetostatic problems is well known[49–51], but has rarely been used in practice due to the complexity of its implementation and an ambiguity associated with gauge freedom. In order to eliminate the gauge freedom and to increase the robustness of the numerical scheme, we used the replacement $|\nabla \times \mathbf{A}_d|^2 \rightarrow |\nabla \mathbf{A}_d|^2$ in the last integral in (2) (see Ref. [52] for details). A well-known consequence of the fact that the vector potential is a continuous field that is noncompact is the Ehrenberg–Siday–Aharonov–Bohm effect. As a result of the noncompactness of $\mathbf{A}_d$, the last integral in (2) is taken over the entirety of three-dimensional space.

We split our implementation into two regions: the interior region—the sample and a relatively small (with respect to the size of the sample) vacuum layer around it, and the exterior region—the rest of the space. The interior was discretized on a regular mesh with spacing 3 nm and the corresponding equations were approximated by a finite-difference method. In the case of periodic boundary conditions in the $xy$ plane, the in-plane mesh size was $256 \times 256$. In the exterior, the field $\mathbf{A}_d$ was approximated by a linear combination of the vector fields $\mathbf{a}_1, \mathbf{a}_2, \ldots$ $\mathbf{a}_n$. The vector fields $\mathbf{a}_i(\mathbf{r})$ were selected to be linearly-independent solutions of the Laplace equation, with regular behavior at infinity. The total number of the $\mathbf{a}_i$ vector field $n$ was chosen to achieve a balance between accuracy and performance. By using toy problems (e.g., uniformly magnetized cuboids), as well as the method of optimal grids[53,54], it was concluded that the most significant factor that limited the overall accuracy of the calculation was the grid spacing in the interior.

For the calculations of the bulk (Fig. 2), we omitted dipole–dipole interactions. When using periodic boundary conditions, we increased the accuracy of the calculations by means of an 8$^{th}$-order finite-difference scheme, which is a generalization of the approach suggested by Donahue and McMichael[55]. In this case, the mesh spacing was set to 0.05 $L_D$ (3.5 nm).

Minimization was carried out using a nonlinear conjugate-gradient method. All of the calculations, including simulations of Lorentz TEM images and phase-shift images, were performed using the high performance GPU-accelerated software *Excalibur*. (see Ref. [56] for more details). For additional verification, a double-check was performed by reproducing some of the solutions for $\mathbf{m}(\mathbf{r})$ using the publicly available software MuMax3[57].

**Derivation of critical fields**. The Hamiltonian in Eq. (2) has an exact solution for a film of finite thickness $t$ (thin bulk) in the presence of a perpendicular applied magnetic field $\mathbf{B}_{\mathrm{ext}}||\hat{\mathbf{e}}_z$:

$$\mathbf{m} = (\sin(\Theta_c)\cos(kz),\ \sin(\Theta_c)\sin(kz),\ \cos(\Theta_c)), \quad (3)$$

$$\mathbf{A}_{\mathrm{d}} = -A_{\mathrm{m}}\big(\cos(kf(z)),\ \sin(kf(z)),\ 0\big), \quad (4)$$

where the cone angle

$$\Theta_c = \arccos\left(\frac{B_{\mathrm{ext}}}{B_{\mathrm{D}} + \mu_0 M_{\mathrm{s}}}\right), \quad (5)$$

the amplitude of the vector potential

$$A_{\mathrm{m}} = \mu_0 M_{\mathrm{s}}\sin(\Theta_c)/k,$$

the auxiliary function

$$f(z) = \max(-t/2, \min(t/2, z)),$$

the wave number $k = \frac{\mathcal{D}}{2\mathcal{A}}$, and $B_{\mathrm{D}} = \frac{\mathcal{D}^2}{2M_{\mathrm{S}}\mathcal{A}}$ is the critical value of the magnetic field in the limit of neglecting demagnetization (this limit corresponds to the case $\mu_0 \to 0$).

In accordance with Eq. (5), we define the critical field $B_c$ to be equal to $B_{\mathrm{D}}$ for a model approach that neglects magnetostatics, while $B_c = B_{\mathrm{D}} + \mu_0 M_{\mathrm{s}}$ for a full model based on Eq. (2). For the above material parameters for FeGe, $B_{\mathrm{D}} = 0.199$ T and $B_{\mathrm{D}} + \mu_0 M_{\mathrm{s}} = 0.682$ T.

**Initial guesses for calculations**. We set the initial guess for the magnetization (the initial magnetization state), representing a vacuum for embedded localized states, to be a perfect conical phase (3). We then use a vortex-like ansatz[7] to embed straight skyrmion strings into this conical background. In some cases, the skyrmion strings in the initial guess are twisted arbitrarily, in order to identify other possible stable states that have different morphologies or twist angles from the spontaneous one. The initial guess for the vector potential field $\mathbf{A}_{\mathrm{d}}$ is always set to be zero.

The positions of the textures in the experimental images were reproduced by adjusting the orientation of $\mathbf{B}_{\mathrm{ext}}$ and using a technique for crafting magnetic textures that is described in Ref. [56] and Supplementary Movie 2. In the simulations, the maximum tilt angle of $\mathbf{B}_{\mathrm{ext}}$ relative to the normal $\hat{\mathbf{e}}_z$ was 1°, which does not exceed the corresponding tolerance of such an angle in the experimental setup.

**Simulation of electron optical-phase shift**. Since our approach for the solution of the micromagnetic problem recovers the magnetic vector potential $\mathbf{A}_{\mathrm{d}}$, simulation of the electron optical-phase shift is straightforward. When the incident electron beam direction is along the negative $z$ axis, the phase shift can be calculated using the following integral[58]:

$$\varphi(x, y) = \frac{2\pi e}{h} \int_{-\infty}^{+\infty} dz\, \mathbf{A}_{\mathrm{d}} \cdot \hat{\mathbf{e}}_z, \quad (6)$$

where $e$ is an elementary (positive) charge ($\sim 1.6 \times 10^{-19}$ C) and $h$ is Planck's constant ($\sim 6.63 \times 10^{-34}$ m$^2$ kg s$^{-1}$).

**Simulation of Lorentz TEM images**. In the phase-object approximation, the wave function of an electron beam[58] transmitted through a sample in the $xy$ plane can be written in the form

$$\Psi_0(x, y) \propto \exp\big(i\varphi(x, y)\big), \quad (7)$$

where $\varphi(x, y)$ is the phase shift defined in Eq. (6). In the Fresnel mode of Lorentz TEM, neglecting aberrations other than defocus, aperture functions, and sources of incoherence and blurring, the wave function at the detector can be written in the form

$$\Psi_{\Delta z}(x, y) \propto \int\int dx'dy'\, \Psi_0(x', y')K(x - x', y - y'), \quad (8)$$

where the kernel

$$K(\xi, \eta) = \exp\left(\frac{i\pi}{\lambda\Delta z}(\xi^2 + \eta^2)\right), \quad (9)$$

the relativistic electron wavelength

$$\lambda = \frac{hc}{\sqrt{(eU)^2 + 2eUm_e c^2}}, \quad (10)$$

$\Delta z$ is the defocus of the imaging lens, $c$ is the speed of light ($\sim 2.99 \times 10^8$ m s$^{-1}$), $U$ is the microscope accelerating voltage, and $m_e$ is the electron rest mass ($\sim 9.11 \times 10^{-31}$ kg).

The integrals in Eq. (8) were calculated using the convolution theorem. The Fourier transform of (7) is straightforward for the case of periodic boundary conditions in the $xy$ plane. For open-boundary conditions in all dimensions, the wave function in Eq. (7) is nonperiodic, and we use a windowed Fourier transform[59]. As a window, we took a rectangular function that is more than an order of magnitude larger than the sample, in order to ensure sufficient accuracy of the Fourier transform. The image intensity was then calculated using the expression

$$I(x, y) \propto |\Psi_{\Delta z}(x, y)|^2. \quad (11)$$

**TEM sample preparation**. TEM samples with well-defined geometries were prepared from a single crystal of B20-type FeGe using a focused ion beam workstation and a lift-out method[60].

**Magnetic imaging in the TEM**. Fresnel defocus imaging and off-axis electron holography were performed in an FEI Titan 60-300 TEM, which was operated at $U = 300$ kV and is equipped with an electron biprism. The microscope was operated in aberration-corrected Lorentz mode with the sample in magnetic-field-free conditions. The conventional microscope objective lens was used to apply the chosen vertical (out-of-plane) magnetic fields to the sample of between $-0.15$ and $+1.5$ T, which were precalibrated using a Hall probe. A liquid-nitrogen-cooled specimen holder (Gatan model 636) was used to vary the specimen temperature between 95 and 380 K. Fresnel defocus images and off-axis electron holograms were recorded using a 4k × 4k Gatan K2 IS direct electron counting detector. The defocus distance was $|\Delta z| = 400\ \mu$m for all images presented in the text.

Multiple off-axis electron holograms, with a 6-s exposure time for each hologram, were recorded to improve the signal-to-noise ratio. The off-axis electron holograms were analyzed using a standard fast Fourier transform algorithm in Holoworks software (Gatan). In order to remove the mean inner potential (MIP) contribution from the total recorded phase shift, phase images were recorded both at low temperature and at room temperature, aligned, and subtracted from each other on the assumption that the MIP contribution is the same and that there are no significant changes in electron-beam-induced specimen charging and dynamical diffraction.

## Data availability

The data that support the findings of this study are available from the corresponding authors upon reasonable request.

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

## Acknowledgements

The authors are grateful to S. Wang, R. Jiang, and T. Denneulin for asistance with TEM sample preparation, and for funding to the European Research Council under the European Union's Horizon 2020 Research and Innovation Programme (Grant No. 856538—project "3D MAGiC"; Grant No. 823717—project "ESTEEM3"; Grant No. 766970—project "Q-SORT"), to the Deutsche Forschungsgemeinschaft (Project-ID 405553726-TRR 270; Priority Programme SPP 2137; Project No. 403502830), and to the DARPA TEE program through grant MIPR# HR0011831554. F.N.R. was supported by Swedish Research Council Grants 642-2013-7837, 2016-06122, and 2018-03659, and by the Göran Gustafsson Foundation for Research in Natural Sciences. N.S.K. acknowledges financial support from the Deutsche Forschungsgemeinschaft through SPP 2137 "Skyrmionics" Grant No. KI 2078/1-1.

## Author contributions

F.Z., F.N.R and N.S.K. conceived the project, and contributed equally to the work. F.Z. and N.S.K. performed TEM experiments and data analysis. F.N.R. performed simulations, with assistance from N.S.K.. All of the authors discussed the results and contributed to the final paper.

## Funding

## Competing interests

The authors declare no competing interests.
