## [Peer Review File · Nature Communications]

Reviewers' Comments:

Reviewer #1:

Remarks to the Author:

In their work, Magnetic Skyrmion Braids, Dr. Kiselev, et al, utilize micromagnetics and electron microscopy to observe what they believe to be skyrmion braids, twisting configurations of skyrmions through the thickness of FeGe lamina. The observation itself has potential to be quite exciting, as complex 3D spin configurations afford new possibilities to both fundamental physics (Topological Hall effects, novel dynamics and resonance properties) as well as for use as basic carriers of information in various spintronic device architectures.

However, there are some questions that should be addressed before I can recommend for publication, as outlined below.

1.) I'm a bit confused of what the actual magnetic configuration is. Clearly there are oscillations in the film normal direction, how is this imprinted on the skyrmion braid? The micromagnetic simulated images do not make this apparent. Particularly, looking at the width of each individual "string" that makes up the braid, there are periodic oscillations through the film thickness. What causes this?

2.) How are these states initialized? Particularly, the nucleation processes seem as though they would be quite complex, and it isn't very clear from the experimental images how they are forming. Could the authors clarify?

3.) I'm impressed with how well the simulated images and the experimental images line up, and this is quite a feat for electron microscopy. However, I would like to see how the details change in a through focal series (both experimental and in theory). This may be useful as a supplementary image, as for large defocus values, information is lost due to the broadening.

4.) In most magnetic images with L-TEM, changing the sense of rotation of the spin-object results in a reversal of contrast, i.e. bright to dark, and the overfocus image ends up looking like the underfocus image of the reversed structure (i.e. see vortices, opposite chirality Bloch skyrmions). Do the authors expect the same for skyrmion braids – in another way of stating this, is the chirality of the skyrmion braid directed related to the contrast, and if the sign of D was reversed, how would the images change.

5.) The authors should briefly a bit more clearly what makes the braid configuration energetically more stable than a straight structure – this seems counterintuitive. Is it a result of wall-wall dipolar interactions?

6.) It may be useful to briefly mention hopfions, another 3D spin texture in magnetic materials with strong DMI that have been recently observed.

Overall, this is an interesting work which has potential to strongly impact the field of skyrmion physics and spin structure in materials with a large DMI. However, the work needs to be a bit better described, especially in terms of the physics. If these aspects are addressed, I do find this work suitable for Nature Communications.

Reviewer #2:

Remarks to the Author:

In the present work, the authors reported the discovery of magnetic skyrmion braids, which represent the twisted rope-like superstructure of magnetic skyrmion strings. The experimental Lorentz transmission electron microscopy images (taken for thin crystal of chiral-lattice magnet FeGe) were compared with the theoretical ones based on the micro-magnetic simulations. The good agreement between the former and latter images evidences the formation magnetic skyrmion braids in the target system.

Recently, magnetic skyrmions are attracting much attention as topologically stable spin texture potentially suitable for information carrier, and the exploration of their three-dimensional structure and dynamics are highly anticipated. The present work nicely demonstrated the formation of brand new superstructure of skyrmion strings, and I think that potentially it deserves the publication in Nature Communications after appropriate revisions.

The followings are my comments and suggestions:

1. In the micro-magnetic simulations, the formation of skyrmion braids was indeed supported for some specific parameter ranges. However, it's not easy to understand why such a structure is energetically favored. It would be helpful for readers if the authors provide more intuitive explanation on the reason for skyrmion braid formation and the magnetic-field-dependence of twisting angle.
2. For bulk crystals of FeGe, clear six-fold magnetic diffraction spots are usually observed in the skyrmion lattice phase, which suggests the absence of such skyrmion braid superstructure. Can the authors discuss the stability of skyrmion braid structure for larger number of skyrmion strings?
3. In the supplementary video 4, skyrmion braids seem to be trapped at the boundary of the sample and suddenly turn into straight skyrmion strings as a function of magnetic field. Is it possible to provide some additional movie corresponding to the situation in Fig. 2 to clarify the detailed deformation process of skyrmion braids? The simulation suggests that the twisting angle is gradually suppressed as a function of magnetic field, and the authors should discuss whether this predicted behavior is consistent with experimental results or not.

May 29, 2021

Manuscript ID: NCOMMS-21-12646-T

Point-by-point response to reviewer reports

We are grateful to both reviewers of our work for constructive comments and that both referees highlight the quality of our work and recommend that our manuscript should be published in Nature Communications. We believe that we have been able to address all of their comments and questions.

For the convenience of the Editor and Reviewers, the changes made in the revised manuscript are listed at the end of the document and also highlighted by red color in the additional document.

Reviewer #1 (Remarks to the Author):

In their work, Magnetic Skyrmion Braids, Dr. Kiselev, et al, utilize micromagnetics and electron microscopy to observe what they believe to be skyrmion braids, twisting configurations of skyrmions through the thickness of FeGe lamina. The observation itself has potential to be quite exciting, as complex 3D spin configurations afford new possibilities to both fundamental physics (Topological Hall effects, novel dynamics and resonance properties) as well as for use as basic carriers of information in various spintronic device architectures.

Response: We greatly appreciate the referee's positive and encouraging evaluation of our work.

However, there are some questions that should be addressed before I can recommend for publication, as outlined below.

1) I'm a bit confused of what the actual magnetic configuration is. Clearly there are oscillations in the film normal direction, how is this imprinted on the skyrmion braid? The micromagnetic simulated images do not make this apparent. Particularly, looking at the width of each individual "string" that makes up the braid, there are periodic oscillations through the film thickness. What causes this?

Response: The magnetic texture that surrounds a skyrmion braid represents a conical spiral. The corresponding modulation in magnetization is described in the following figure, which is taken from our earlier publication (New J. Phys. **18** (2016), 045002).

The modulation is visualized in the inset to Fig. 1a and is depicted in color at the boundaries of the boxes in Figs 1a-d. As the applied magnetic field is increased, the conical angle decreases towards zero and the cone phase tends to a field-polarized state. This behavior is reflected in a change in color at the boundaries of the boxes towards white, which corresponds to the spins pointing in the applied field direction (*i.e.*, the z axis).

The bumps on the isosurfaces ($m_z = 0$) that are used to visualize the skyrmion strings, which Reviewer 1 refers to as “periodic oscillations through the film thickness”, are associated with the modulations in magnetization of the conical phase. The appearance of such bumps on skyrmion isosurfaces is known (see, *e.g.*, Supplementary Movie 1 in F. N. Rybakov et al., Phys. Rev. Lett. **115** (2015), 117201 and A. O. Leonov et al., J. Phys.: Condens. Matter **28** (2016), 35LT01). The following figure, which is taken from our earlier publication (Phys. Rev. Lett. **120** (2018), 197203), shows the same effect for two interacting skyrmion strings in the presence of a large applied magnetic field.

It should be noted that the contours ($m_z = \text{constant}$) are not axisymmetric for skyrmions embedded in a conical phase, as would be the case for skyrmions embedded in the ferromagnetic state. The “amplitude” of the bumps on the isosurface is proportional to the cone angle and becomes less pronounced as the applied magnetic field is increased. The number of bumps is equal to the number of periods of the conical spiral along the z axis. In order to clarify this behavior to the reader, we have added the following sentences to the caption to Fig. 1:

“The presence of bumps on the isosurfaces results from the magnetization modulations in the surrounding conical phase. These bumps become less pronounced with increasing B_{ext} as the cone phase approaches a field-polarized state.”

2) How are these states initialized? Particularly, the nucleation processes seem as though they would be quite complex, and it isn't very clear from the experimental images how they are forming. Could the authors clarify?

Response: We have included a recipe for the nucleation of skyrmion braids at the bottom of Page 4:

“For experimental observations of skyrmion braids, we designed the following protocol...”

This protocol is based on the following steps:

- A few skyrmions can be introduced into the sample by reversing the direction of the applied magnetic field and changing its magnitude. The skyrmions are typically attached to the edges of the sample at low fields.
- According to our earlier study (Phys. Rev. Lett. **120** (2018), 197203), in order to “detach” the skyrmions from the edges, a relatively strong magnetic field needs to be applied to suppress the edge modulations and to “push” the skyrmions towards the center of the sample. This step is required to avoid interactions between the skyrmions and the edge modulations.
- When the skyrmions form a cluster near the center of the sample, the applied magnetic field is decreased gradually to make the skyrmion cluster unstable, with respect to a transition to a braided superstructure.

In order to clarify the above steps, we have reproduced them experimentally and included them in the form of a **new Supplementary Video 5**.

3) I'm impressed with how well the simulated images and the experimental images line up, and this is quite a feat for electron microscopy. However, I would like to see how the details change in a through focal series (both experimental and in theory). This may be useful as a supplementary image, as for large defocus values, information is lost due to the broadening.

Response: Lorentz TEM contrast is indeed sensitive to defocus and can contain misleading and non-intuitive features. We have therefore recorded a through-focal series of Lorentz TEM images of skyrmion braids composed of 3 skyrmion strings and performed corresponding theoretical calculations. The results are included as a **new Supplementary Fig. 10**.

4) In most magnetic images with L-TEM, changing the sense of rotation of the spin-object results in a reversal of contrast, i.e. bright to dark, and the overfocus image ends up looking like the underfocus image of the reversed structure (i.e. see vortices, opposite chirality Bloch skyrmions). Do the authors expect the same for skyrmion braids – in another way of stating

this, is the chirality of the skyrmion braid directed related to the contrast, and if the sign of D was reversed, how would the images change.

Response: To a first approximation, Lorentz TEM contrast is sensitive to the in-plane magnetic induction projected along the electron beam direction. The rotation of the magnetic induction in a skyrmion string therefore determines the Lorentz TEM contrast. We expect the same behavior for skyrmion braids, as they are composed of skyrmions. If the sign of the DMI interaction constant is reserved, then the rotation of the skyrmions is also reserved and the contrast should reverse accordingly. In order to illustrate this behavior, we have included simulations for negative DMI as a **new Supplementary Fig. 11**.

5) The authors should briefly a bit more clearly what makes the braid configuration energetically more stable than a straight structure – this seems counterintuitive. Is it a result of wall-wall dipolar interactions?

Response:

Numerical studies have shown that dipole-dipole interactions and the sample geometry are not a self-sufficient cause of braiding. In particular, wall-wall dipolar interactions are not the cause of stability.

We performed a series of calculations for periodic boundary conditions and omitted dipole-dipole interactions. For different mesh densities, energy minimization was performed for three states: (i) a pure cone phase; (ii) a cluster of 6 straight skyrmion strings embedded in the cone phase; (iii) a braid of 6 skyrmion strings embedded in the cone phase. Representative results for states (ii) and (iii) are shown in Fig. 2a in the revised manuscript. Since an analytical solution for the cone phase is known, one can estimate the accuracy of the numerical method. When refining the mesh, we observed convergence between the analytical and numerical solutions for the cone phase, as shown in the red curve in the figure below, confirming the accuracy of the numerical calculations.

A positive difference between the energies of the straight skyrmion cluster and the skyrmion braid (see the green line in the plots below) provides evidence that the skyrmion braid is energetically favorable. The energy difference (green line) is several orders of magnitude greater than the numerical error (red line).

In order to clarify the mechanism of skyrmion braid stability, we have added a figure to the manuscript to illustrate the gain in energy terms for twisted skyrmions compared to straight skyrmion strings. We have also added the following text to the manuscript:

“The mechanism of skyrmion braid stability therefore results from a sophisticated interplay of multiple energy terms and depends on the geometry of the system. The dipole-dipole interaction does not play a fundamental role in the energy stability of a braid, but contributes to it. Instead, the stability of a braid results primarily from the energies of exchange-type interactions associated with texture distortions.”

6) It may be useful to briefly mention hopfions, another 3D spin texture in magnetic materials with strong DMI that have been recently observed.

Response: We have added the following statement about hopfions to the Discussion:

“In contrast to magnetic hopfions^{46–48}, skyrmion braids cannot be localized completely in three dimensions without singularities. In other words, skyrmion strings start on one surface of a sample and end on the other surface.”

We have also added the following references:

46. Liu, Y., Lake, R. K., Zang, J. Binding a hopfion in a chiral magnetic nanodisk. Phys. Rev. B **98**, 174437 (2018).
47. Voinescu, R. et al. Hopf solitons in helical and conical backgrounds of chiral magnetic solids. Phys. Rev. Lett. **125**, 057201 (2020).
48. Kent, N. et al. Creation and observation of Hopfions in magnetic multilayer systems. Nat. Commun. **12**, 1562 (2021).

7) Overall, this is an interesting work which has potential to strongly impact the field of skyrmion physics and spin structure in materials with a large DMI. However, the work needs to be a bit better described, especially in terms of the physics. If these aspects are addressed, I do find this work suitable for Nature Communications.

Response: We appreciate the positive and encouraging evaluation of our work. A complete list of changes is provided at the end of this document.

Reviewer #2 (Remarks to the Author):

In the present work, the authors reported the discovery of magnetic skyrmion braids, which represent the twisted rope-like superstructure of magnetic skyrmion strings. The experimental Lorentz transmission electron microscopy images (taken for thin crystal of chiral-lattice magnet FeGe) were compared with the theoretical ones based on the micro-magnetic simulations. The good agreement between the former and latter images evidences the formation magnetic skyrmion braids in the target system.

Recently, magnetic skyrmions are attracting much attention as topologically stable spin texture potentially suitable for information carrier, and the exploration of their three-dimensional structure and dynamics are highly anticipated. The present work nicely demonstrated the formation of brand new superstructure of skyrmion strings, and I think that potentially it deserves the publication in Nature Communications after appropriate revisions.

Response: We greatly appreciate the positive and encouraging evaluation of our work and have included a point-by-point response to the referee's comments below.

The followings are my comments and suggestions:

1. In the micro-magnetic simulations, the formation of skyrmion braids was indeed supported for some specific parameter ranges. However, it's not easy to understand why such a structure is energetically favored. It would be helpful for readers if the authors provide more intuitive explanation on the reason for skyrmion braid formation and the magnetic-field-dependence of twisting angle.

Response: A similar request was made by Reviewer 1. We hope that our explanation and revision, which includes an additional figure (Fig. 2), clarifies the mechanism of skyrmion braid stability.

2. For bulk crystals of FeGe, clear six-fold magnetic diffraction spots are usually observed in the skyrmion lattice phase, which suggests the absence of such skyrmion braid superstructure. Can the authors discuss the stability of skyrmion braid structure for larger number of skyrmion strings?

Response:

Skyrmion braids can contain larger numbers of strings. We have added the following text to the manuscript:

“See additional examples of superstructures of braided skyrmions in Supplementary Figs 12 and 13.”

A new Supplementary Fig. 12 (see below) shows experimental observations of skyrmion braids that contain more than 7 skyrmion strings.

A new Supplementary Fig. 13 (see below) shows an example of the formation of large superstructures from braided skyrmions.

In neutron scattering experiments, six-fold magnetic diffraction spots have been recorded from bulk FeGe close to the Curie temperature [E. Moskvina et al., Phys. Rev. Lett. **110** (2013), 077207]. In contrast, our experimental and theoretical analysis are carried out in the low temperature régime, which can be described more closely by micromagnetics according to Brown's terminology. The study of braiding in the presence of stronger thermal fluctuations and their influence on diffraction patterns are interesting problems that are beyond the scope of the present work.

3. In Supplementary Video 4, skyrmion braids seem to be trapped at the boundary of the sample and suddenly turn into straight skyrmion strings as a function of magnetic field. Is it possible to provide some additional movie corresponding to the situation in Fig. 2 to clarify the detailed deformation process of skyrmion braids? The simulation suggests that the twisting angle is gradually suppressed as a function of magnetic field, and the authors should discuss whether this predicted behavior is consistent with experimental results or not.

Response: We repeated the experiment for a larger sample, based on the protocol described in the manuscript. The results are shown in **Supplementary Videos 5 and 6**, which illustrate reproducibility of skyrmion braids with increasing and decreasing applied magnetic field. In the videos, a cluster of skyrmions in the middle of the domain is not affected by edge modulations. Supplementary Videos 1 and 6 confirm consistency between theory and experiment, respectively, in the suppression of twisting by an increasing applied magnetic field.

List of major changes in the revised manuscript:

1. We added a non-referenced abstract according to Nature Communications guidelines and rephrased the introductory paragraphs for consistency.
2. A new Fig. 2 and a new paragraph on page 3 have been added to describe the stability mechanism of skyrmion braids.
3. Figure 1h has been moved to a new Fig. 2.
4. A new paragraph at the end of the manuscript and 3 additional references have been added to discuss hopfions.
5. Supplementary Videos 5 and 6 have been added to describe the nucleation mechanism of the braids, as well as braiding and unbraiding effects.
6. Supplementary Fig. 10 has been added to describe the variation of Lorentz TEM contrast as a function of defocus.
7. Supplementary Fig. 11 has been added to illustrate Lorentz TEM contrast of skyrmion braids for crystals of opposite chirality (*i.e.*, opposite sign of DMI constant).
8. Supplementary Fig. 12 has been added to show experimental observations of skyrmion braids composed of more than 7 skyrmion strings.
9. Supplementary Fig. 13 has been added to show an example of how braided skyrmions can form large superstructures.

Reviewers' Comments:

Reviewer #1:

Remarks to the Author:

In their response, the authors have clearly and systematically addressed all of my comments. As such, I am much more confident in their work, and it is presented in a much clearer and convincing manner, with additional details also allowing for much easier reproducibility. Their observations of clearly new 3D spin textures in magnetic materials with DMI is both timely and novel. As such, I am not happy to recommend for publication.

Reviewer #2:

Remarks to the Author:

I have read through the revised manuscript, and found that the authors appropriately responded to my previous concerns. I think that the present work would contribute to the deeper understanding of three-dimensional superstructure of this unique topological object, and recommend its publication in the present form.